# Neuroprotective and Disease-Modifying Effects of the Triazinetrione ACD856, a Positive Allosteric Modulator of Trk-Receptors for the Treatment of Cognitive Dysfunction in Alzheimer’s Disease

**DOI:** 10.3390/ijms241311159

**Published:** 2023-07-06

**Authors:** Cristina Parrado Fernandez, Sanja Juric, Maria Backlund, Märta Dahlström, Nather Madjid, Veronica Lidell, Azita Rasti, Johan Sandin, Gunnar Nordvall, Pontus Forsell

**Affiliations:** 1AlzeCure Pharma AB, Hälsovägen 7, 141 57 Huddinge, Sweden; cristina.parrado@alzecurepharma.com (C.P.F.); sanja.juric@alzecurepharma.com (S.J.); maria.backlund@alzecurepharma.com (M.B.); marta.dahlstrom@alzecurepharma.com (M.D.); veronica.lidell@alzecurepharma.com (V.L.);; 2Division of Neuroscience, Care and Society, Department of Neurogeriatrics, Karolinska Institutet, 171 77 Solna, Sweden

**Keywords:** brain-derived neurotrophic factor, nerve growth factor, Alzheimer’s disease, allosteric modulator, cognitive function, anti-depressant

## Abstract

The introduction of anti-amyloid monoclonal antibodies against Alzheimer’s disease (AD) is of high importance. However, even though treated patients show very little amyloid pathology, there is only a modest effect on the rate of cognitive decline. Although this effect can possibly increase over time, there is still a need for alternative treatments that will improve cognitive function in patients with AD. Therefore, the purpose of this study was to characterize the triazinetrione ACD856, a novel pan-Trk positive allosteric modulator, in multiple models to address its neuroprotective and potential disease-modifying effects. The pharmacological effect of ACD856 was tested in recombinant cell lines, primary cortical neurons, or animals. We demonstrate that ACD856 enhanced NGF-induced neurite outgrowth, increased the levels of the pre-synaptic protein SNAP25 in PC12 cells, and increased the degree of phosphorylated TrkB in SH-SY5Y cells. In primary cortical neurons, ACD856 led to increased levels of phospho-ERK1/2, showed a neuroprotective effect against amyloid-beta or energy-deprivation-induced neurotoxicity, and increased the levels of brain-derived neurotrophic factor (BDNF). Consequently, administration of ACD856 resulted in a significant increase in BDNF in the brains of 21 months old mice. Furthermore, repeated administration of ACD856 resulted in a sustained anti-depressant effect, which lasted up to seven days, suggesting effects that go beyond merely symptomatic effects. In conclusion, the results confirm ACD856 as a cognitive enhancer, but more importantly, they provide substantial in vitro and in vivo evidence of neuroprotective and long-term effects that contribute to neurotrophic support and increased neuroplasticity. Presumably, the described effects of ACD856 may improve cognition, increase resilience, and promote neurorestorative processes, thereby leading to a healthier brain in patients with AD.

## 1. Introduction

Neurotrophins are a well-studied class of neurotrophic proteins that play important roles in both neurological and non-neurological functions. The initial discovery of nerve growth factor (NGF) [1] paved the way for the identification of homolog proteins, including brain-derived neurotrophic factor (BDNF) [2], neurotrophin (NT)-3 [3], and NT-4 [4]. The first tropomyosin receptor kinase (Trk) was originally discovered as a receptor tyrosine kinase with transforming growth activity [5]. It was later demonstrated that neurotrophins bind to the cognate Trk family of receptors where NGF preferentially binds to TrkA [6], BDNF and NT-4 bind to TrkB [7], and NT-3 binds to TrkC [8].

The p75 neurotrophin receptor (p75NTR) binds the pro-neurotrophins with higher affinity than the neurotrophins [9]. Additionally, the findings of the formation of TrkA/P75NTR heterodimers [10], the binding of NT-3 to TrkA [11], and differential signaling [12] are likely to contribute to the complex signaling of Trk-receptors.

Subsequently, the roles of both NGF and BDNF and their receptors in AD were clarified [13,14,15]. In the past 10 years, the role of BDNF-Val66Met (BDNF-Met) polymorphism in the progression of cognitive decline in both sporadic and familial AD has been well described [16,17,18]. Additionally, those who carry both the BDNF-Met and the ApoE4 alleles have significantly higher amyloid-beta load than non-carriers [19,20]. Studies with ApoE4 and BDNF-Met allele carriers have demonstrated a correlation between genotype and memory impairment in older non-symptomatic ApoE4 carriers [21,22], in patients with mild cognitive impairment [23], and in preclinical AD [22]. Also, the BDNF-Met allele is associated with increased levels of biomarkers such as tau and phospho-tau and poorer cognitive performance in non-symptomatic as well as in symptomatic dominantly inherited Alzheimer’s disease [24].

Apart from its role in AD, BDNF has also been shown to be decreased in neuropsychiatric disorders such as depression [25] and post-traumatic stress disorder (PTSD) [26]. A role for BDNF in Rett syndrome has also been described, suggesting that dysregulation of the BDNF gene may affect the severity of this pathology [27,28,29].

The results from later-stage clinical trials with the two approved anti-amyloid monoclonal antibodies, aducanumab and lecanemab [30], as well as the more recently described and not yet approved donanemab [31], have only demonstrated a 35% reduction in deterioration of the integrated Alzheimer’s disease rating scale (iARDS) over time. This, in combination with the established role of BDNF-Met in cognitive impairment in AD, clearly shows that there is a need for new treatments for AD that improve memory using a different mechanism of action than the anti-amyloid antibodies. One such approach could be to target the BDNF pathway. Considering that most AD patients or their care partners ranked improved memory of higher importance than clearance of amyloid pathology [32], and in light of the high societal costs associated with anti-amyloid monoclonal antibodies, it is crucial to develop cost-effective therapies with cognitive enhancing effects. Indeed, several attempts have been made to identify small molecule activators of neurotrophin signaling [33,34] or inducers of neurotrophin levels [35]. Additionally, E2511, a selective positive modulator of TrkA, was reported to specifically enhance trophic signaling and reverse cholinergic neuronal loss in a model of AD [36]. Importantly, these effects were observed without the induction of hyperalgesia, suggesting that it is possible to differentiate NGF-induced trophic signaling from the induction of hyperalgesia. We recently described the identification of triazinetrione derivatives, including ACD856, a pan-Trk positive allosteric modulator with a significant pro-cognitive effect [37]. Additionally, ACD856 increased the levels of certain neurotransmitters and demonstrated an antidepressant-like function [38]. ACD856 has completed both a single and multiple ascending dose study in healthy volunteers. The compound demonstrated excellent pharmacokinetic properties and was well tolerated [39].

We describe herein the preclinical characterization of ACD856 and demonstrate that the compound has significant neuroprotective effects, increases the levels of BDNF, potentiates neurite outgrowth, and has evidence of neuroplastic adaption leading to improved cognitive function and long-lasting antidepressant-like effects.

## 2. Results

### 2.1. In Vitro Characterization of ACD856

We have previously demonstrated that ACD856 binds to Trk-receptors, increases the catalytic efficiency of the kinase activity, and, thereby, leads to increased cellular signaling [37]. In this study, we have utilized cells recombinantly overexpressing Trk-receptors, cell lines with endogenous expression, primary neurons, or in vivo studies to further addressed the mechanism by which triazinetriones increase the activity of Trk-receptors and downstream signaling. We and others have used the PathHunter assay to demonstrate pharmacological effects on Trk-receptors [37]. The assay measures the interaction between SHC-1 and phosphorylated Y490 on TrkA-receptors. As can be seen from Figure 1a, U2OS/TrkA/SHC1/p75NTR cells were treated with increasing concentrations of NGF. Those cells exposed to seven different concentrations of ACD856 (3 nM to 30 mM) enhanced in a dose-dependent manner the activity of TrkA well above the activity that was seen with only NGF, suggesting an increased interaction of SHC-1 with phosphorylated Y490 of TrkA. To elucidate if ACD856 increased Trk-signaling by improving the ligand-receptor interaction, we investigated if ACD856 had an effect on the interaction between NGF and TrkA. Figure 1b demonstrates that ACD856 increases the activity of TrkA without changing the affinity of NGF towards TrkA. There were no differences in the EC_50_ values of NGF or ACD856 that clearly could be linked to a change in either the concentrations of the compound or NGF. The mean EC_50_-values were 4.3 +/− 0.6 ng/mL and 195 +/− 21 nM (mean +/− SD, *n* = 8 or 6) for NGF and ACD856, respectively. These results clearly demonstrate that the effects of ACD856 on enhanced Trk-signaling are not due to changes in the affinity between ligand and Trk-receptor. Additionally, an activation of approximately 60% of TrkA by ACD856 could be observed in the absence of NGF (Figure 1a,b), suggesting a partial agonistic modulatory effect in the absence of a ligand. Whether or not this basal activation is a consequence of activated receptors due to highly overexpressed recombinant receptors remains to be determined in cells with endogenously expressed Trk-receptors. In summary, ACD856 enhances the phosphorylation state of Y490 of Trk-receptors and has a pharmacological profile resembling that of a positive efficacy modulator with partial agonism.

### 2.2. ACD856 Increased BDNF-Induced Phosphorylation of TrkB Receptors and Promoted Activation of the BDNF/ERK Signaling Pathway

Having established that the effects of ACD856 on Trk-signaling are not due to increased ligand-receptor interaction but rather due to the increased phosphorylation state of the receptor, we further characterized the molecular mechanisms by which ACD856 enhances Trk-signaling.

First, we analyzed whether the compound increased tyrosine phosphorylation of the TrkB receptor in the human neuroblastoma-derived cell line SH-SY5Y overexpressing TrkB. BDNF could, in a dose- and time-dependent manner, increase TrkB phosphorylation as determined by the use of a phospho-TkB ELISA (Figure 2a). The same methodology was used to demonstrate that treatment of cells with 300 nM ACD856 and BDNF demonstrated a significant increase in the phosphorylation of TrkB receptors (Figure 2b). Interestingly, there was a trend towards bell-shaped effects since higher concentrations of ACD856 did not lead to the same level of phospho-TrkB as 300 nM ACD856. A trend to bell-shaped effects can also be seen in Figure 1b, where the highest concentrations of ACD856 rendered a smaller effect than the lower concentrations of ACD856. The bell-shaped effect, or the lack of dose–response effect seen in Figure 2b, could be explained by a negative feedback loop in overexpressing stable cell lines in which a high degree of receptor activation initiates dephosphorylation/deactivation of the Trk-receptors. This effect seems to be restricted to short-time experiments studying the phosphorylation state of the Trk-receptors, as exemplified in Figure 1b and Figure 2b. Longer timepoints with functional readouts do not show this effect, as exemplified in Figure 3.

Second, we addressed if ACD856 could enhance signaling downstream of Trk-receptors. One of the main pathways downstream of Trk-receptors is the extracellular regulated kinase 1/2 (ERK1/2) pathway. This, in combination with the fact that BDNF-TrkB signaling through phosphorylation of ERK1/2 mediates synaptic plasticity and memory formation [40], prompted us to investigate whether ACD856 could enhance the effect of BDNF on the ERK1/2 signaling cascade in primary neurons. Because ERK1/2 phosphorylation may be induced by the binding of not only neurotrophins but also other growth factors to their respective tyrosine kinase receptors, we first assessed the effect of various growth factors or neurotrophins on ERK1/2 phosphorylation in cortical neurons using Western blot. The results (Figure 2c) indicate that among all growth factors analyzed, BDNF and NT3 were the most effective growth factors with respect to inducing phosphorylation of ERK1/2 in primary cortical neurons. Additionally, BDNF and NT3 were the only growth factors that induced phosphorylation of Trk-receptors as judged by the use of an anti-phospho-Trk antibody (pY674/675). The effect of ACD856 on the phosphorylation of ERK1/2 was then assessed using immunocytochemistry in cortical neurons in the absence of exogenous BDNF to identify the subcellular distribution of phosphor-ERK1/2 and to identify differences in phospho-ERK1/2 in cell bodies and neurites. As shown in Figure 2d, levels of phospho-ERK1/2 increased significantly in neurites after ACD856 treatment. The number of phospho-ERK1/2 positive neurites (Figure 2d) and the total fluorescence intensity of phospho-ERK1/2 positive neurites (Appendix A) both increased significantly. However, the cell bodies, which were the most intensely stained regions (Figure 2e), did not demonstrate any significant difference in phospho-ERK1/2 content (Appendix A). The fact that significant changes were observed in the neurites but not in the cell body implies that the ERK cascade is involved in the mechanism of action of ACD856 with the potential to regulate BDNF’s actions on neurites and synaptic function.

### 2.3. Neurite Outgrowth-Promoting Activity of ACD856 in NGF-Stimulated PC12 Cells

To assess the functional activity of ACD856, we used a phenotypic high-content assay to study the effects of ACD856 on NGF-induced proliferation and differentiation of PC12 cells. Briefly, PC12 cells were incubated with DMSO or increasing amounts of ACD856 in the presence of 3 ng NGF/mL for 5 days. As can be seen in Figure 3, ACD856 enhanced NGF-induced neurite outgrowth, both as measured by neurite total length per neuron (Figure 3a) and by neurite total length per well (Figure 3b). ACD856 did not only promote neurite extension but also significantly increased the number of neuritic processes as determined by the total number of neurites per neuron (Figure 3c). Interestingly, ACD856 had no effect on the proliferation of PC12 cells in the presence of 3 ng NGF/mL (Figure 3d).

The synaptosome-associated protein 25 kDa (SNAP-25) is a pre-synaptic protein involved in exocytosis and neurotransmitter release. Since synaptic dysfunction is an early hallmark of AD, pharmacological means to compensate for the reduced expression of synaptic proteins could be a way forward to normalize the expression of synaptic proteins, including SNAP25. Its expression and localization in PC12 cells have previously been described [41], and hence, the expression of SNAP25 was analyzed in NGF-treated PC12 cells in the presence of increasing concentrations of ACD856. Figure 4 demonstrates a trend to dose-dependent increase in the number of SNAP25-positive neurites, which became significant at the highest concentration tested. As can be seen in Figure 4b, SNAP25 in NGF-stimulated PC12 is localized mainly to the cell body, neurites, and buddings of the neurites resembling immature dendritic spines.

### 2.4. ACD856 Increases BDNF Levels In Vitro and In Vivo 

Studies in cortical neurons have demonstrated an autoregulatory-positive feedback loop of BDNF [42,43,44] that promotes local synaptic plasticity and consolidates hippocampal long-term memory in vivo [45]. These findings suggest that BDNF-mediated neuroplasticity may require auto-upregulation of BDNF gene expression.

Consequently, we aimed to investigate whether the presence of ACD856 could result in an increased production of BDNF in primary cortical neurons.

Neurons were incubated with increasing concentrations of ACD856, and the levels of BDNF were thereafter determined. As demonstrated in Figure 5, ACD856 increased the levels of BDNF protein by approximately 30%, with a significant increase observed at 100 nM of ACD856 compared to DMSO-treated cells. Interestingly, the levels of BDNF in DMSO-treated cells was 7.9 ng/mL, which is in line with previously reported EC_50_-values for BDNF binding to TrkB [37].

Since BDNF has been shown to be reduced in the hippocampus and parietal cortex in patients with AD [46], we sought to determine the effects of ACD856 on BDNF levels in these regions in mice. Consequently, 3 months old mice were administered 3 mg/kg ACD856 by oral administration once daily for five days. Thereafter, the hippocampus and cortex were dissected and homogenized, and BDNF levels were determined. To ensure that the animals were administered a dose that would have a pharmacological effect, we performed a forced swim test to address the antidepressant-like effects that we previously reported for ACD856 [37]. As can be seen in Figure 6, the administered dose resulted in a significant effect on both the latency to the first immobility and the immobility time in the forced swim test. However, ACD856 had no effect on BDNF levels in the cortex (Figure 6a) or in the hippocampus (Figure 6b) after 5 days of repeated administration, even though the compound demonstrated a significant antidepressant-like effect (Figure 6c,d). Our conclusion from this experiment is that short-term administration in young animals does not lead to increased levels of BDNF in the investigated areas of the brain. Whether or not this is due to the short-term exposure to ACD856 or to the fact that the compound has no effect on BDNF levels in a naïve, young, healthy animal remains to be determined.

There is ample evidence suggesting that aging correlates to reduced levels of BDNF in different regions of the brain. Thus, we also sought to determine the effect of ACD856 on BDNF levels in aged animals by using brains from old mice. Based upon the lack of previous studies on a minimal pharmacologically effective dose of ACD856 in aged animals, we chose a somewhat higher dose than the one used in young animals. Thus, 21 months old mice were administered 5 mg/kg of ACD856 once daily for 4 weeks by subcutaneous injection. As can be seen in Figure 7, old animals that were administered ACD856 demonstrated a significant increase in BDNF levels when homogenate from the entire left-brain hemisphere was analyzed for BDNF levels. In contrast to the results on BDNF levels obtained with young animals, the BDNF pathway might be reduced or, in other ways, compromised in aged animals, which makes the old animals more amenable to treatment with ACD856 than young naïve animals, at least with respect to increasing the levels of BDNF.

### 2.5. Neuroprotective Role of ACD856 in Energy-Deprived Neuronal Cells

Mitochondrial and metabolic alterations in the brains of cognitive-impaired animals are associated with disruption of excitatory and inhibitory synaptic activity and neurotrophic factor release. Furthermore, neocortical and hippocampal glucose hypometabolism anticipates a cognitive decline in normal aging and Alzheimer’s disease. To test whether ACD856 and neurotrophins have a neuroprotective role when energy metabolism is deficient, we used primary neurons cultured after the withdrawal of glucose and pyruvate and with only glutamine as an energy source, which serves as an alternate substrate in Kreb’s cycle. To prevent glutamate excitotoxicity, the glutamine concentration was maintained at 2 mM, which is comparable to the amount found in a healthy brain [47]. First, we addressed the metabolic activity of the cells by measuring the NADH-mediated reduction in resazurin. In comparison to untreated cells, cells stimulated with 10 ng/mL BDNF or NGF showed significantly higher fluorescence, indicating increased metabolic activity (Figure 8a). By using a multiplexed assay, we simultaneously investigated cell membrane integrity and ATP production. As shown in Figure 8b, primary neurons deprived of glucose and pyruvate were able to increase ATP levels when cultured with BDNF or NGF. In addition, ACD856 demonstrated a dose-dependent increase in ATP, which was significant at 1, 3, and 10 µM (Figure 8c). As can be seen in Figure 8d,e, NGF, BDNF, or ACD856 also improved cell membrane integrity. Future studies addressing the effects of ACD856 on mitochondrial function by measuring oxygen respiration are currently ongoing and will be reported elsewhere.

### 2.6. ACD856 Prevents Aβ_1-42_-Induced Synaptotoxicity in Cortical Neurons

BDNF has previously been shown to protect against amyloid-beta (Aβ) induced neurotoxicity in vitro and in vivo [48], and BDNF-peptidomimetics has been shown to protect against Aβ-induced neurotoxicity by increasing BDNF levels [49]. Given that synaptic dysfunction is one of the earliest functional hallmarks of AD correlating well to cognitive dysfunction and that several reports show that Aβ_1-42_ reduces the levels of the SNAP-25, we investigated the effect of Aβ_1-42_ induced synaptotoxicity in cortical neurons. Cells were exposed to 10 μM Aβ_1-42_ for 96 h with or without ACD856, and the levels of SNAP-25 in neurites were determined by immunocytochemistry. The neurites demonstrated punctuated and dystrophic morphology after treatment with Aβ_1-42_ (Figure 9a,b). After quantification, the number of SNAP25-positive neurites was significantly reduced after treatment with Aβ_1-42_ (Figure 9c). ACD856 could significantly and almost fully protect the cortical neurons from Aβ_1-42_-induced toxicity, with a trend to protective effect at 300 nM, which was significant at 1 μM (Figure 9d).

### 2.7. Pharmacokinetics of ACD856

To address the exposure of ACD856 in animals, pharmacokinetic (PK) studies were performed in mice, rats, and minipigs. The former addresses pharmacodynamic relationships, and the latter two as species for GLP-compliant toxicokinetic evaluation. The PK results in mice demonstrated an elimination half-life of 1.4 h after subcutaneous administration (s.c.) of 1 mg/kg of ACD856. This dose resulted in a total concentration in plasma (C_max_) of 3550 nM (1375 ng/mL) and a total brain exposure of 402 nM (162.7 ng/mL) when analyzed 30 min after administration. Additionally, oral administration of 5 mg/kg of ACD856 yielded an elimination half-life of 3.3 h with a plasma and brain concentration of 65,960 nM (25,500 ng/mL) and 1350 nM (523 ng/mL), respectively, when samples were taken 60 min after administration. Administration via subcutaneous injection or oral administration led to approximately the same exposure in the brain when adjusted for the administered dose. The plasma protein binding of ACD856 was measured in plasma from CD-1 or C57/Bl6 mice, and it was shown to be 96.2–98.2%. Thus, the plasma concentration of unbound ACD856 30 min after administration of 1.0 mg/kg (s.c.) would be 65–134 nM (25–52 ng/mL), depending on the mice strain.

### 2.8. In Vivo Behavioral Studies

ACD856 and a structurally similar compound from this chemical class, ACD855, have shown potent pro-cognitive effects in various cognition models such as passive avoidance, novel object recognition, and Morris water maze [37] after subcutaneous administrations at doses ranging from 0.3 mg/kg up to 3 mg/kg.

In the present studies, we used the passive avoidance task to assess the effect of repeated administration of ACD856 prior to cognitive testing. Animals were administered ACD856 subcutaneously once daily for 4 days prior to the behavioral task. Results show that the minimal effective dose of ACD856 was lowered from 0.3 mg/kg after a single dose to 0.1 mg/kg after repeated administration (Figure 10a). This is not the result of compound accumulation, given the half-life of approximately 1.4 h after subcutaneous administration in mice. These results indicate that repeated administration of ACD856 caused neuroplastic adaptive changes that further enhanced its activity in vivo, which could be linked to the reported acute vs. long-term effects of BDNF [50].

Given the data from the passive avoidance model, we wanted to explore whether the effects after repeated administration could be mediated by long-term neuroplastic changes that could potentially also outlast the elimination of the compound.

To investigate this, we conducted experiments in the forced swim test, a model to assess antidepressant-like activity in rodents. Herein, we administered ACD856 once daily for 5 days and then waited either 1, 3, or 7 days after the last administration before conducting in vivo behavioral testing. Results showed that the compound was able to significantly reduce the immobility in the test, even up to 7 days after the last administration (Figure 10b). It is interesting to note that BDNF and activation of TrkB can induce increased plasticity [51,52].

## 3. Discussion

Patients with AD exhibit a large degeneration of cholinergic neurons in specific areas of the brain, such as the basal forebrain [53]. The role of cholinergic dysfunction in AD and age-related cognitive dysfunction was proposed early on [54]. This hypothesis has been supported by the demonstration that NGF prevents the degeneration of cholinergic neurons in the basal forebrain in non-human primates [55]. Several attempts have been made to deliver NGF centrally [56,57], and there are indeed results demonstrating a positive effect of NGF [58,59]. Also, acetylcholine esterase inhibitors have been shown to have a long-term effect on both mortality and cognitive decline [60]. More recently, it was demonstrated that a reduction in NGF levels leads to astrocytic conversion and induction of a neurotoxic response [61], suggesting that improved NGF levels or TrkA-signaling in astrocytes can have a neuroprotective effect. Taken together, these data demonstrate the important role of NGF in the maintenance of cholinergic function. Hence, therapies aiming at enhancing the NGF/TrkA cascade could potentially improve or restore cholinergic function.

BDNF is well known to play a role in synaptic plasticity [62,63], cognition, regeneration of neurons [64], and protective effects against several toxins, including Ab-induced synaptotoxicity [65] or tau-induced neurodegeneration [66]. Therefore, there is a great potential for therapeutics that will enhance BDNF/TrkB signaling in a wide range of different diseases, including Alzheimer’s disease [67].

The in vitro experiments presented herein indicate that ACD856 functioned as a positive allosteric modulator with partial agonism. Furthermore, the molecule possessed a neurotrophic efficacy, as judged by its positive effects on neurite outgrowth and SNAP25 levels in PC12 cells. Additionally, the effects observed after glucose and pyruvate withdrawal in primary neurons suggest a neuroprotective role for ACD856, which causes neurons to rapidly adapt to maintain high metabolic activity and an increased pool of ATP. Consistent with this, our results also indicate that ACD856 may help neurons switch to a glutamine-fueled mitochondrial metabolism which may reduce excitotoxic stress but also provide them with a considerable amount of energy that is especially important for brain regions affected by hypoglycemia. Even more interestingly, the neuroprotective effect of ACD856 was not limited to hypoglycemic situations since also a protective effect was observed of ACD856 in Ab_1-42_-induced synaptotoxicity. This demonstrates that the compound could have a broad range of neuroprotective effects, mimicking some of the previously reported effects of BDNF.

BDNF was early on demonstrated to be involved in learning processes by the use of heterozygous BDNF-knockout mice [68,69]. Furthermore, it was shown that BDNF levels were reduced in aged animals, which correlated with reduced hippocampal volume and memory performance. BDNF expression was reported to be selectively reduced in the ventral tegmental area (VTA) in aged mice [70], demonstrating that aging indeed can influence BDNF levels. The role of the BDNF Val66Met polymorphism has also been demonstrated in older adults where carriers demonstrate reduced executive function [71] and episodic memory [72]. The finding that ACD856 increased the levels of BDNF in both cortical neurons and in the brain of aged animals (Figure 5 and Figure 7) could be a likely explanation for the cognitive enhancement and antidepressant-like effects of ACD856 observed in vivo. The increased levels of BDNF after treatment with ACD856 demonstrate that the compound also provides neurotrophic support in preclinical models by increasing the levels of BDNF.

Thus, the neurotropic support, and the neuroprotective and neurorestorative effects described in this paper, along with the lack of major adverse events in the clinical studies, including pain or allodynia [39], suggest that ACD856 is well suited as a therapy to enhance, protect and possibly restore cognitive domains in the brain. The fact that ACD856 can increase the activity of TrkA, TrkB, and TrkC receptors, thus enhancing the signaling of all the neurotrophins, combined with its high exposure in the CNS, makes it likely that ACD856 will have both restorative and protective effects as well as a cognitive enhancing effect in diseases characterized by cognitive dysfunction. The lowering of the effective dose and the sustained antidepressant-like effects seen in vivo after repeated dosing of ACD856 suggest a mechanism involving neuronal adaptation.

Estimations of human efficacious plasma concentration and pharmacologically active dose (PAD) were based on the estimated unbound C_max_ and the observed pharmacological effect in the mouse cognition passive avoidance model after single-dose administration. The minimum and maximal pharmacologically active dose (PAD) in mice after single administrations were identified as 0.3 mg/kg (s.c.) and 1.0 mg/kg (s.c.), respectively. Since the in vitro TrkA EC_50_ value was demonstrated to be similar between humans and rats, and the homology of TrkA between rats and mice is high (96% identity) [37], it is believed that the PAD for ACD856 corresponds to a human dose generating an unbound C_max_ in the range of 50 to 70 ng/mL. Interestingly, a single dose of 30 mg ACD856 generated a plasma concentration of unbound ACD856 of 32 ng/mL in men [39], which is well in line with the values of unbound ACD856 in mice plasma at 1.0 mg/kg reported herein (25–52 ng/mL).

The introduction of new treatments for AD, including the anti-amyloid monoclonal antibodies aducanumab, lecanemab (aka Leqembi), and donanemab opens up new possibilities for adjuvant therapies. The preclinical data presented in this article suggests that ACD856 may fulfill many of the criteria for adjuvant therapy to anti-amyloid treatment or as a stand-alone treatment if the preclinical results are reproduced in patients.

## 4. Materials and Methods

### 4.1. Reagents and Chemicals

Recombinant human NGF (450-01) and BDNF (450-02) were purchased from PeproTech (London, UK). GlutaMAX™ Supplement (35050061), StemPro™ Accutase™ Cell Dissociation Reagent (11599686), heat-inactivated fetal bovine serum (FBS) (10500064) or horse serum (26050070), G418 sulfate (10131027), penicillin-streptomycin (15140148), protease inhibitor (Pierce™ A32955) and phosphatase inhibitor (A32957), modified Dulbecco’s PBS Tween™ buffer (Pierce™ 28346), MEM (31095029), Neurobasal SFM (21103049), Neurobasal™-A medium, no D-glucose, no sodium pyruvate (A2477501), RPMI Medium 1640 (11875101), Leibovitz’s L-15 medium (11415049), B-27™ supplement (17504-044), EBSS (14155063) were purchased from Thermo Fisher Scientific (Waltham, MA, USA). Resazurin (ready-to-use solution) was acquired from TCI Europe (Zwijndrecht, Belgium). The mitochondrial ToxGlo™ assay kit (G8001) was purchased from Promega Corporation (Madison, WI, USA). The PathHunter Detection Kit (93-0001) was obtained from Eurofins DiscoverX.

The following antibodies were used in this study: mouse anti-beta tubulin (G-8) (sc-55529, Santa Cruz, CA, USA), rabbit anti-phospho-p44/42 MAPK (ERK1/2) (Thr202/Tyr204) (D13.14.4E) (4370, Cell Signaling), rabbit anti-SNAP25 (PA1-740, Invitrogen, Fisher Scientific, Gothenburg, Sweden), rabbit phospho-TrkA (Tyr674/675)/TrkB (Tyr706/707) (C50F3) (4621, Cell Signaling), and mouse anti-beta actin (C4) (sc-47778, Santa Cruz, CA, USA).

### 4.2. Cell Lines and Cell Culture Conditions

The human osteosarcoma cell lines U2OS-TrkA/p75-SHC1 and U2OS-TrkB/p75-SHC1 were acquired from Eurofins DiscoverX Corporation (Fremont, CA, USA). Cells were maintained subconfluent in MEM medium supplemented with 10% FBS, 1% penicillin/streptomycin, and hygromycin/geneticin/puromycin. Immortalized SH-SY5Y cell line expressing TrkB was purchased from Kerafast (Boston, MA, USA) and maintained in complete growth medium, consisting of RPMI1640, GlutaMAX, 10% FBS, 1% penicillin/streptomycin, and 0.3 mg/mL geneticin. The PC12 cell line was purchased from ATCC (LGC Standards GmbH, Wesel, Germany) and maintained in DMEM supplemented with 10% FBS and 1% penicillin/streptomycin. All cell cultures were maintained at 37 °C in a humidified atmosphere with 5% CO_2_.

### 4.3. Animals

All animal experiments were conducted in C57BL/6J male mice from Charles River, Germany. All animals were permitted to acclimate to the animal facilities and had unrestricted access to food and water. Animals were handled by the same researcher for at least three days before the start of the studies. All protocols were approved by the Stockholm County court’s regional ethics committee in compliance with the Swedish Animal Welfare Act 2018:1192 and directive 2010/63/EU (NIH publication No. 85-23). The experiments were carried out under the ethical permit ID1640 (in vivo studies) and 2181-2021 (primary neuronal cell cultures). Every effort was made to reduce animal suffering and the number of animals used in the experiments.

### 4.4. Isolation and Culture of Mouse Primary Neurons

Mouse cortical neuronal cultures were prepared from day 16–17 of embryonic development. For precise identification, the embryonic brain cortex was dissected in 5-mL EBBS (Ca^2+^/Mg^2+^ free) on a 60 mm Petri dish using a Leica stereo microscope. The tissue was transferred to a 15 mL tube and digested for 15 min at 37 °C in EBSS with trypsin-EDTA (0.25%). Once the tissue had sunk to the bottom, the solution was discarded. Thereafter, cells were mechanically disaggregated in EBSS containing DNAse I using a glass Pasteur pipette. The supernatant was transferred to a fresh 15 mL tube, and the remaining non-dissociated tissue was discarded. Trituration was performed twice, and the pooled supernatant was centrifuged at 500 rpm for 5 min. Cell pellets were then resuspended in 10 mL of fresh Neurobasal medium containing 2% B-27 and 0.5 mM L-glutamine and 100 units/mL penicillin and streptomycin. Ten microliters of the diluted cells were placed in a hemocytometer for counting. Dissociated primary cortical neurons were cultured in poly-D-lysine coated 96-well plates at a cell density as described in assay sections below. Half of the medium was changed every 5 days. All cell cultures were maintained at 37 °C in a humidified atmosphere with 5% CO_2_.

### 4.5. TrkA and TrkB PathHunter^®^ Cell-Based Assays

The TrkA or TrkB cell-based PathHunter^®^ assays were used to characterize the potency and efficacy of ACD856. The assays use enzyme fragment complementation (EFC) to analyze protein–protein interactions. TrkA or TrkB receptors are phosphorylated upon ligand activation (NFG and BDNF, respectively), leading to an interaction with SHC-1. Trk-receptors are fused to an enzyme donor tag, and SHC-1 is fused to a larger part of β-gal called an enzyme acceptor. The interaction leads to an active β-gal enzyme that will hydrolyze a substrate generating a luminescence signal. The assays were performed as previously described (#Cell paper). Briefly, 10,000 cells per well in 20 mL of MEM medium supplemented with 0.5% horse serum were seeded in 384-well white plates (CulturePlate, PerkinElmer, Waltham, MA, USA) and incubated overnight at 37 °C in a humidified atmosphere with 5% CO_2_. Thereafter, in vitro effect of ACD856 was tested using eight different concentrations ranging from 0.01 to 30 µM using 10 ng/mL of NGF or BDNF. The test compound plate was diluted with 10 mL of Leibovitz’s L15 medium containing 10 ng/mL of NGF or BDNF and pipetted into the cell plate using a Biomek NX automated workstation. Plates also included min-control and max-control wells (containing assay media or 30 ng/mL of ligand, respectively). After a three-hour incubation at room temperature, 7 mL of substrate was added to each well, followed by a 60 min incubation. The chemiluminescence was then measured using a SpectraMax software (version 7.0.3) and a iD5 plate reader from Molecular Devices. Raw values were used to calculate EC_50_ values (µM) and the efficacy (%) of ACD856 by setting the maximal response (DMSO control + ligand 30 ng/mL) to 100% and the negative control (DMSO control without ligand) to 0%.

### 4.6. Resazurin Assay

Resazurin assay was used to measure the metabolic capacity of PCN cells treated with increasing doses of BDNF, NGF, and ACD856 using a model of energy deprivation-induced neurotoxicity. Cells were seeded at 10 × 10^5^ cells/well in 96-well plates and treated at DIV4–5 in Neurobasal-A media with 2% B27 and 2 mM glutamine (no glucose or pyruvate was present). Two hours after treatment, a 10% volume of resazurin ready-to-use solution was added to the cell culture media, and the plates were returned to the incubator for a further four hours. Resazurin reduction is directly proportional to metabolic function and was evaluated fluorometrically at 540 nm excitation and 580 nm emission wavelengths using a SpectraMax iD5 plate reader. Results were presented as a percentage change compared to the control group.

### 4.7. Measurement of Cell Membrane Integrity and ATP Levels

Mitochondrial ToxGlo assay was used to evaluate the cell membrane integrity and the ATP levels of PCN cells. At DIV4–5, cells seeded at 10 × 10^5^ cells/well in 96-well plates were treated for six and three hours with increasing doses of neurotrophins and ACD856, respectively, in Neurobasal-A media containing 2% B27 and 2 mM glutamine. No glucose or pyruvate was present in the cell media to prevent non-mitochondrial ATP production from glycolysis. Then, bis-AAF-R110, a fluorogenic substrate that is exclusively cleaved by proteases released from membrane-compromised cells, was added, and the plate was incubated at 37 °C for 30 min. Fluorescence was measured at 485 nm (Ex)/530 nm (Em) using a SpectraMax iD5 plate reader. Next, the ATP detection reagent was added, and luminescence was measured after 5 min incubation to assess ATP levels in lysed cells. The results were displayed as a percentage change relative to the control group.

### 4.8. Measurement of Human Phospho-TrkB by ELISA 

A DuoSet ELISA human phospho-TrkB was used to measure tyrosine-phosphorylated human TrkB in cell lysates. SH-SY5Y-TrkB cells (10 × 10^4^ cells) were seeded in 96-well plates overnight. The day after, cells were exposed for 10 min to 30 ng/mL of BDNF +/− ACD856 at a concentration of 0.3–3 mM and immediately lysed on ice for 15 min, followed by centrifugation at 500× *g* for 5 min. The ELISA procedure was performed following the manufacturer’s specifications. Briefly, cell supernatant was transferred to an ELISA plate previously coated with the capture antibody and blocked at RT for 2 h (samples were run in duplicate). Then, the plate was incubated for 2 h with the diluted HRP-labelled anti-phosphotyrosine antibody, followed by 20 min of incubation with substrate Solution. The reaction was stopped with 2 M H_2_SO_4_ and immediately measured using a microplate reader set to 450 nm and with wavelength correction set to 540 nm. Intermediate washing steps were performed before and after each incubation step with a wash buffer.

### 4.9. Quantification of BDNF by ELISA

Detection of endogenous BDNF was performed with a Human/Mouse BDNF DuoSet ELISA (DY248) in PCN (10 × 10^4^ cells/well) between days in vitro (DIV) 10–14. The cells were microscopically examined before and after treatments, and wells with abnormal neuron morphology or disrupted cell layer were excluded. The determination was carried out according to the manufacturer’s specifications. Briefly, 96-well Maxisorp plates (NUNC) were coated with 100 mL/well of capture antibody and incubated overnight. Non-specific binding was blocked for 90 min using 5% BSA in tris-buffered saline with 0.3% Triton X-100 (TBS-T), followed by repeated rinsing with TBS-T. After 6 h of treatment, 10× lysis buffer containing protease inhibitor and EDTA-free was added to the cell plates and incubated for 20 min, followed by centrifugation at 1000 rpm for 2 min. Then, 100 mL of samples or standard samples were added to the coated plate and kept overnight at 4 °C. ELISA plates were washed four times, followed by an incubation of 2 h with 100 mL/well of detection antibody. After repeated washing steps, 100 mL/well of Streptavidin-HRP was added and incubated for 20 min. After washing, 100 mL/well of substrate solution was added and incubated for 25 min. The reaction was stopped by adding 50 mL of stop solution, and the absorbance was immediately measured at 450 nm using a SpectraMax iD5 plate reader from Molecular Devices. Unless otherwise mentioned, all incubations were conducted in a shaker at 300 rpm and room temperature. All samples were run in duplicates. The standard curve ranged from 11 to 1500 pg/mL BDNF. Concentrations of BDNF were calculated using the regression line of the BDNF standard curve. According to the manufacturer, the BDNF ELISA kit has no cross-reactivity with other neurotrophic factors, such as NGF, NT-3, and NT-4, at 50 ng/mL.

### 4.10. Immunocytochemistry Analysis of Neurite Outgrowth, SNAP-25, and ERK1/2 Staining

PC12 cells (2000 cells/well) were cultured in 384-well black poly-D-lysine plates coated with a clear bottom (PerkinElmer ViewPlate), coated with 0.1 mg/mL Collagen IV and 0.025 mg/mL vitronectin, and treated with 0 or 3 ng/mL NGF, for 5 days in a humidified atmosphere at 5% CO_2_. Mouse primary cortical neurons (25 × 10^3^ cells per well) were cultured in 96-well plates coated with poly-D-lysine and incubated with DMSO or ACD856. Incubations were terminated by the addition of 16% paraformaldehyde to yield a final concentration of 4%. Thereafter, plates were placed at +4 °C for at least 60 min. Fixed cells were washed twice with TBS-T and blocked for 1 h in TBS-T containing 2.5% BSA. Thereafter, sequential incubations with primary and secondary antibodies in blocking buffer were performed at 4 °C overnight. PC12 cells were stained with anti-β-tubulin mouse antibody (Santa Cruz sc-55529) (dilution 1:200) and Alexa fluor 488-goat anti-mouse antibody (dilution 1:750). Anti-SNAP25 antibody (dilution 1:200) and Alexa Fluor 647-goat anti-rabbit antibody (dilution 1:750) were used for staining of SNAP25. Mouse primary cortical neurons were stained with anti-phospho-p44/42 MAPK (ERK1/2) rabbit monoclonal antibody (Cell Signaling CS-4370) (dilution 1:200) and Alexa Fluor 647 goat anti-rabbit antibody (dilution 1:750). Nuclei were stained with Hoechst fluorescent DNA dye. High-content imaging/analysis was performed on a Thermo Scientific Cellomics Array Scan VTI HCS Reader using Cellomics Scan Software (ver 6.6.1).

### 4.11. Western Blot Analysis

Phosphorylated ERK 1/2 (pERK1/2) levels were evaluated in mouse primary neurons between DIV10–15. Cells were seeded at (60 × 10^4^ cells) in 6-well plates coated with poly-D-Lysine. The activation of the Trk signaling pathway was induced with 10 ng/mL ligand for 20 min. Cells were thereafter immediately lysed on ice, sonicated, and denatured at 95 °C for 10 min on a heating block. Proteins were separated on a Novex Bis-Tris 4–12% gel and transferred to a PVDF membrane using an iBlot instrument (Invitrogen). The membrane was blocked using 5% BSA in TBS-T followed by incubation with anti-phospho-p44/42 MAPK (ERK1/2) rabbit monoclonal antibody (Cell Signaling CS-4370) (dilution 1:1000) in TBS-T with 0.1% BSA for at least 60 min. Membranes were thereafter washed three times and thereafter incubated with horseradish peroxidase-coupled secondary goat anti-rabbit antibodies for 60 min. After washing, a Western blot detection reagent (Forte, Millipore) was added, and immunoreactive proteins were detected by luminescence using C-DiGit Blot Scanner and quantified with Image Studio™ Software (version 5.2). (LI-COR Biosciences, Bad Homburg, Germany).

### 4.12. Passive Avoidance Test

The passive avoidance (PA) task is an associate learning paradigm that is based on Pavlovian fear-conditioning and instrumental conditioning and conducted as previously described [37] in male C57BL/6J mice (*n* = 7–8/group) from Charles River Laboratories, Sulzfeld, Germany. Briefly, vehicle or ACD856 (0.1, 0.3, or 1 mg/kg) were administered either as a single subcutaneous injection on the training day (Day 4) or once daily for 4 days (Day 1–4). The last injection was given 60 min prior to training. Moreover, on the training day (Day 4), scopolamine (Sigma Aldrich, Stockholm, Sweden) at 0.3 mg/kg or vehicle was administered once s.c. 30 min prior to training. No compound was administered on the test day, conducted 24 h after training (Day 5), where retention latencies and time spent in the bright compartment were determined.

### 4.13. Forced Swim Test

The FST is one of the most frequently used behavioral tests for measuring depressive-like behavior in rodents. Animals placed in cylinders containing water rapidly become immobile, demonstrated by floating passively or making only movements necessary to remain afloat. Based on an immobility response induced by inescapable exposure to stress, the FST also has strong predictive validity because short-term administration of antidepressant compounds from a variety of pharmacological classes reduces immobility time in the FST. These drugs include tricyclic antidepressants, MAO inhibitors, atypical antidepressants, and SSRIs [73].

Depression-like behavior was assessed in male C57BL/6J mice using a modified version of the FST, as described previously [74]. This included a two-day protocol test, with pre-exposure to water 24 h prior to the test (day one of FST), which seems to be a more accurate and sensitive technique in detecting depression-like behavior in mice than the standard method of a single exposure (one day test) [75,76]. Animals were individually placed in a vertical glass cylinder (50 cm high, 20 cm in diameter, CMA) filled with tap water up to 35 cm (25 ± 0.5 °C). Two swimming sessions were conducted: a 10 min pre-test (day one) followed 24 h later (day two) by a 6 min test session. The total duration of immobility, as well as latency to the first immobility, were recorded during the 6 min test. After each swimming session, the mice were gently removed and placed in a home cage together with dry napkins. Immobility was defined as floating passively in an upright position in water with only small movements necessary to keep the head above the water’s surface. The floating time was considered an index of depression-like behavior. The glass cylinder was cleaned thoroughly between each animal.

### 4.14. Plasma Protein Binding

The fraction unbound drug (fu) in plasma from mice was determined by equilibrium dialysis at 37 °C for 4 h using RED devices. The drug molecule at a concentration of 10 µM was added to plasma from CD-1 or C57/Bl6 mice and dialyzed against phosphate-buffered saline (pH 7.4). After dialysis, the drug concentration in the buffer and plasma was quantified by LC-MS/MS analysis, as described below.

### 4.15. Pharmacokinetic Studies and LC-MS/MS Analysis of ACD856

Male C57BL/6J mice were administered with a single subcutaneous dose of 1 mg/kg ACD856, and blood was collected at 7 timepoints, i.e., at 15 min, 30 min, 1 h, 2 h, 4 h, 8 h, and 24 h post-dosing. Plasma was separated within 30 min by centrifugation at 3000× *g* (+4 °C) for 10 min. Pharmacokinetic parameters were evaluated by non-compartmental analysis based on plasma compound concentrations determined by LC-MS/MS analysis.

Standards for a calibration curve, ranging from 10 to 10,000 nM, and quality control (QC) samples were prepared in blank mouse plasma by spiking plasma with ACD856 from a 10 mM DMSO stock. Calibration standards and QC samples were prepared just before the analysis. Plasma samples were precipitated by 100% methanol containing 50 nM Warfarin as an internal standard. After centrifugation, the supernatant was transferred to an autosampler glass vial, and the concentration of ACD856 was determined by LC-MS/MS. An Acquity UPLC system (Waters, Solna, Sweden) coupled to a triple quadrupole mass spectrometer (Xevo TQ-S micro, Waters, Solna, Sweden) operating in multiple reaction monitoring (MRM) modes were used for the detection of the compound. Samples were separated on a BEH C18 column (2.1 × 50 mm, 1.7 µm, Waters), kept at 60 °C using a gradient consisting of a mixture of (A) 0.1% formic acid in water and (B) 0.1% formic acid in acetonitrile, as follows: 0–0.4 min, 5% B; 1.4–1.6 min, 100% B; 1.65–2.00 min, 5% B. Flow rate was 0.5 mL/min and the samples were kept at 10 °C in an autosampler. For ionization of compounds, electrospray in negative mode (ESI-) was used with the following conditions: capillary voltage, 1.9 kV; desolvation gas temperature, 500 °C; gas flow, 1000 L/hr. The optimized transitions for ACD856 were precursor ion 385.968 m/z, product ions 160.994 and 267.087 m/z, and cone and collision energies were 30 V and 8 V, respectively. TargetLynx software (version 4.2) was used for the integration of compound peak area, and a calibration curve was made by fitting the analyte concentrations versus the peak area ratios of the analyte to internal standard using the most suitable regression model and weighting method.

### 4.16. Statistical Analysis

All statistical computations were carried out using GraphPad Prism Software, version 9.3.1. Half maximal effective concentrations (EC50 values) were calculated using non-linear regression analysis. Unpaired t-tests were used to compare the two groups. For experiments with more than two groups, a one-way analysis of variance (ANOVA) was performed, followed by Dunnett’s post hoc tests for groupwise comparisons. Data values are expressed as mean ± standard error of the mean (SEM) unless otherwise stated in the legend of each figure. The significance level was accepted at *p* < 0.05 for a 95% confidence interval. 

## Figures and Tables

**Figure 1 ijms-24-11159-f001:**
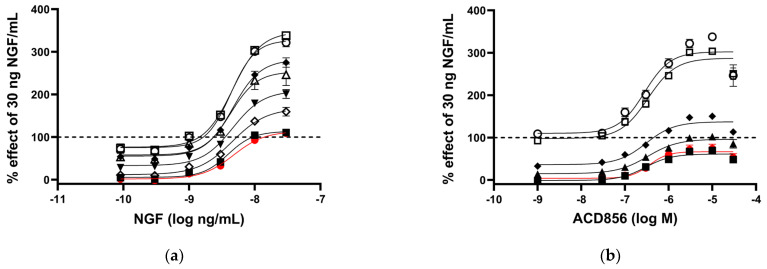
Increased activation of NGF-TrkA signaling through interaction of SHC−1 and TrkA. U2OS−TrkA/SHC−1/p75NTR cells were treated with DMSO, NGF, or NGF plus ACD856 for 3 h. The response was measured with luminescence as described in Materials and Methods. The same data were used to visualize the results either as a consequence of a change in the concentrations of NGF or as a change in the concentrations of ACD856. Dose–response curves of NGF in the presence of increasing concentrations of ACD856 (**a**). Red circle and line depict activity in the absence of NGF. The following concentrations of ACD856 were used: 3 nM (solid squares), 100 nm (open diamonds), 300 nM (solid triangles), 1 mM (solid diamonds), 3 mM (open circle), 10 mM (open squares), and 30 mM (open triangles.) Dose–response curves of ACD856 in the absence or presence of increasing concentrations of NGF (**b**). Red circle and line depict activity in the absence of NGF. The following concentrations of NGF were used: 0.3 ng/mL (solid squares), 1 ng/mL (solid triangles), 3 ng/mL (solid diamonds), 10 ng/mL (open squares), and 30 ng/mL (open circle). Dashed line in both figures indicates the activity of 30 ng NGF/mL in the presence of DMSO, to which all other values were normalized to. Data shown are the mean +/− SEM (*n* = 12) from three independent experiments where each experiment was performed in quadruplicates.

**Figure 2 ijms-24-11159-f002:**
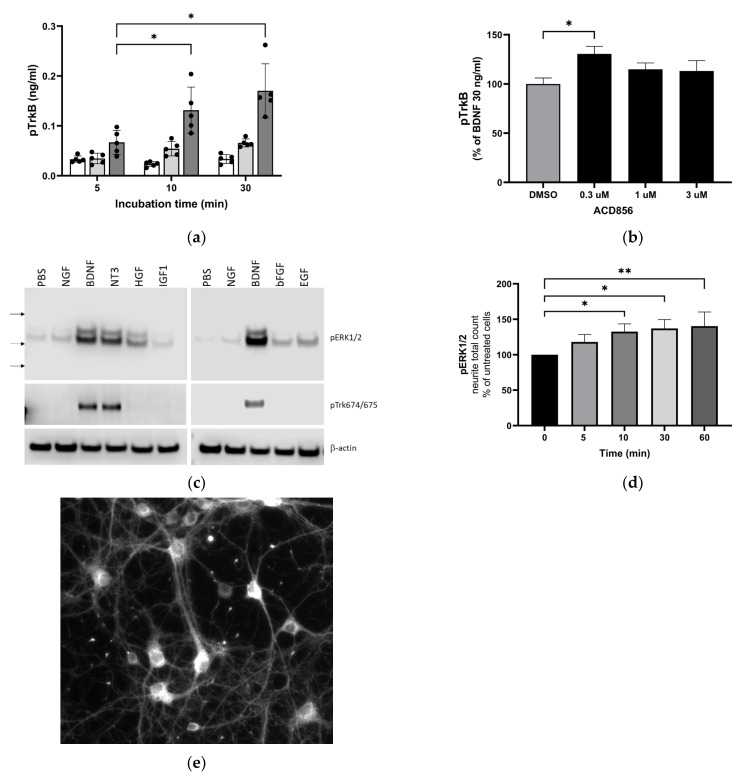
Increased activation of TrkB and downstream signaling through ERKK1/2 MAPK phosphorylation by BDNF and ACD856. Levels of phosphorylated TrkB receptors were determined by ELISA in SH-SY5Y cells overexpressing the human TrkB receptor (**a**). Effect of 0 (open bars), 10 (light grey bars), or 30 ng BDNF/mL (dark grey bars) and time course on BDNF-induced TrkB tyrosine phosphorylation are depicted as the mean of one experiment performed in quadruplicates +/− SEM * *p* < 0.05. SH-SY5Y-TrkB cells were incubated with 30 ng BDNF/mL for 10 min in the absence (DMSO) or presence of increasing concentrations of ACD856 (**b**). A significant increase in human pTrkB was observed with 300 nM ACD856. Data shown are the mean value +/− SEM from three independent experiments where each experiment was performed using at least 5 technical repeats. The effect of different neurotrophic factors on the phosphorylation of ERK1/2 and Trk-receptors was investigated (**c**). Cortical neurons were stimulated with 10 ng/mL of the indicated factors, and the levels of pERK1/2 (upper panel), phospho-Trk (tyrosine 674/675) (mid panel), and beta-actin as a loading control (lower panel) were analyzed by Western blot. Primary cortical neurons were incubated with 300 nM ACD856 for the indicated time periods in the absence of exogenous BDNF (**d**). The levels of phosphorylated ERK1/2 were determined by immunocytochemistry, and the total count per well of pERK1/2-positive neurites was determined. Fluorescence image at ×10 magnification showing phospho-ERK1/2 detected with immunocytochemistry in cortical neurons (**e**). * *p* < 0.05, ** *p* < 0.01 compared to control group at one-way ANOVA with Dunnett’s multiple comparisons test. Data shown are the mean value of all replicates +/− SEM from three different experiments.

**Figure 3 ijms-24-11159-f003:**
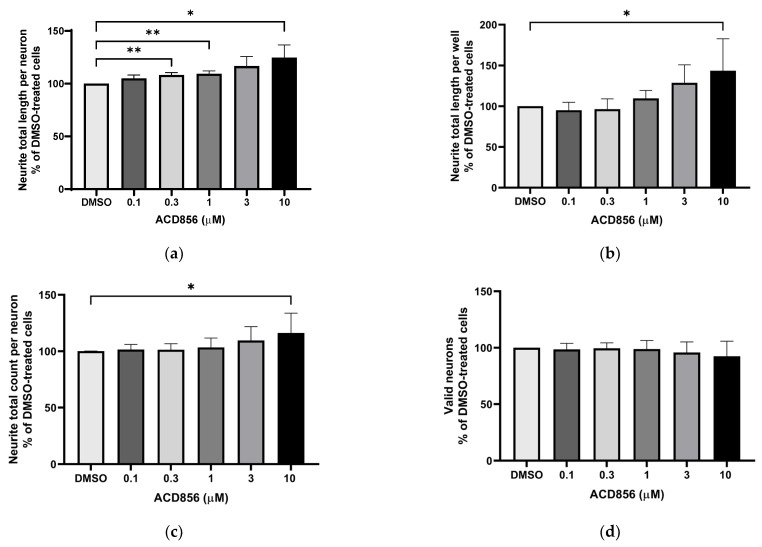
Potentiation of NGF-induced neurite outgrowth in PC12 cells by ACD856. PC12 cells were treated with 3 ng NGF/mL for 5 days in the absence or presence of increasing concentrations of ACD856. Cells were fixed with PFA, and anti-tubulin antibodies were used to visualize neurite processes. A dose–response effect of ACD856 was observed for neurite total length per neuron (**a**), neurite total length per well (**b**), and neurite total count per neuron (**c**). There was no significant effect on the number of neurons (**d**). * *p*-value  <  0.05, ** *p*-value  <  0.01 compared to control group at one-way ANOVA with Dunnett’s multiple comparisons test. Data shown are the mean of average values from each experiment +/− SD (*n* = 5 independent experiments).

**Figure 4 ijms-24-11159-f004:**
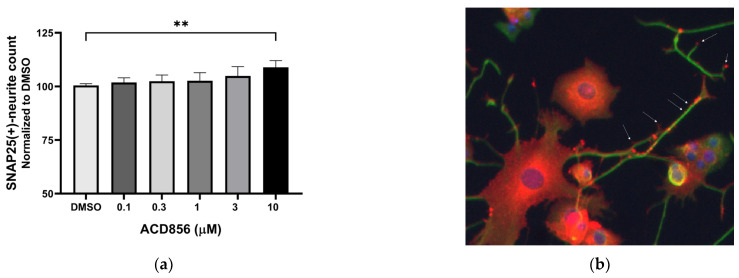
Increase in SNAP25-containing neurites in PC12 cells by ACD856. PC12 cells were treated with 3 ng NGF/mL for 5 days in the absence or presence of increasing concentrations of ACD856. Cells were fixed with PFA, and anti-SNAP25 (red) or anti-beta tubulin (green) antibodies were used to visualize cells and neurites. A dose–response effect of ACD856 was observed for SNAP25-positive neurite total count per neuron (**a**). The SNAP25-protein was mainly localized to the cell body, neurites, and especially in buddings of neurites or at nerve endings (indicated with arrows). Image was acquired with ×20 magnification. (**b**). ** *p*-value  <  0.01, compared to control group at one-way ANOVA with Dunnett’s multiple comparisons test. Data shown are the mean of average values from individual experiments +/− SD (*n* = 3).

**Figure 5 ijms-24-11159-f005:**
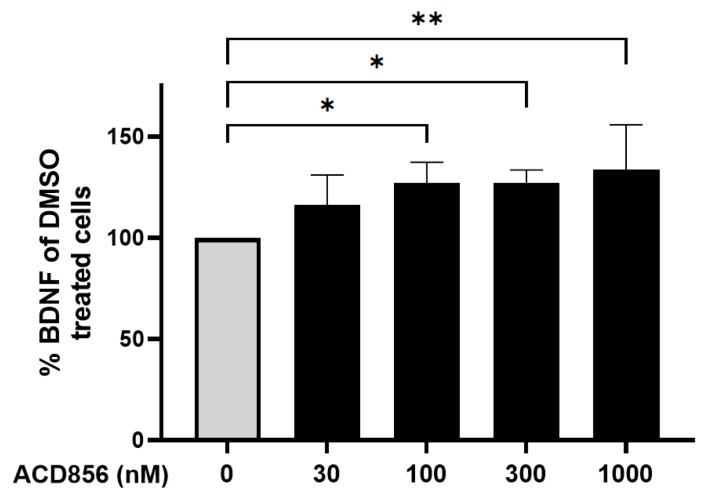
Cortical neurons were incubated with ACD856 in NB-media for 6 h, and the BDNF levels were determined by ELISA. Data shown are the mean +/− SD from four independent experiments where each experiment was performed using six technical replicates. Each sample was analyzed in duplicates in the ELISA assay. * *p* < 0.05, ** *p* < 0.01 vs. DMSO-control values. The absolute level of BDNF in DMSO-treated samples was 7.9 +/− 1.7 ng/mL (mean +/− SD, *n* = 5).

**Figure 6 ijms-24-11159-f006:**
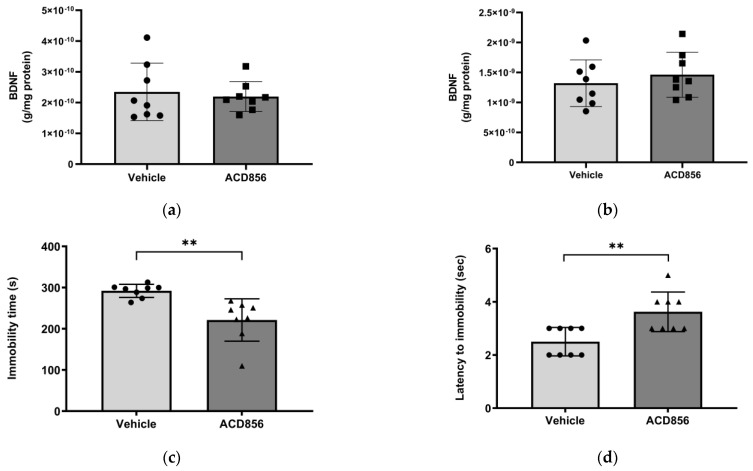
Effects of ACD856 on BDNF levels and depression-like behavior in mice. Animals were dosed with 3 mg/kg ACD856 by oral administration once daily for 5 days. Cortex (**a**) and hippocampus (**b**) were dissected, and the BDNF levels were determined by ELISA after completion of the forced swim test (**c**,**d**). The immobility time (**c**) and the latency to first immobility (**d**) were recorded manually. Data shown are the means +/− SD, *n* = 8. ** *p* < 0.01 vs. vehicle-treated animals. Light grey bars and black circles indicates vehicle treated animals whereas as dark grey bars and black squares or triangles indicated ACD856 treated animals.

**Figure 7 ijms-24-11159-f007:**
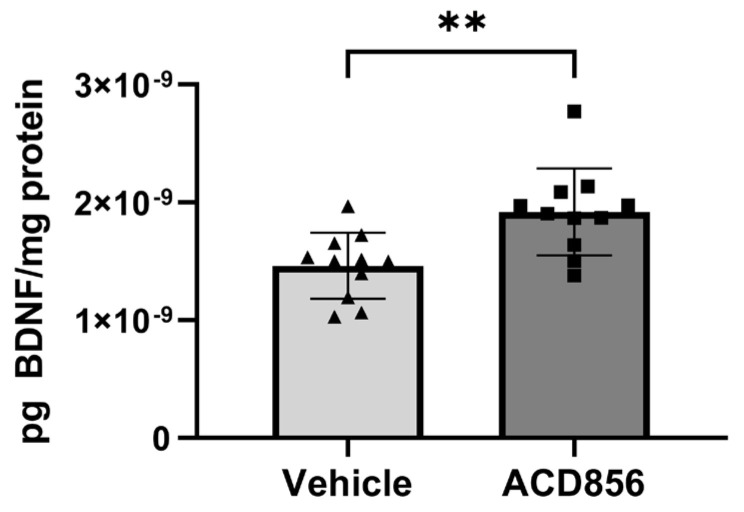
ACD856 increases the levels of BDNF in the brain of aged animals. Twenty-one months old mice were dosed with 5 mg/kg ACD856 once daily for 4 weeks by s.c. injection. The left hemisphere of each brain was homogenized, and the BDNF levels were determined by ELISA. Data shown are the mean +/− SD, *n* = 11–12 animals. All samples were analyzed in duplicates in the ELISA assay, and BDNF levels were normalized to protein content in each sample. Individual data for each animal is the means of duplicate determinations. ** *p* < 0.01 vs. vehicle-treated animals using Students *t*-test. Light grey bars and black triangles indicates vehicle treated animals whereas as dark grey bars and black squares indicated ACD856 treated animals.

**Figure 8 ijms-24-11159-f008:**
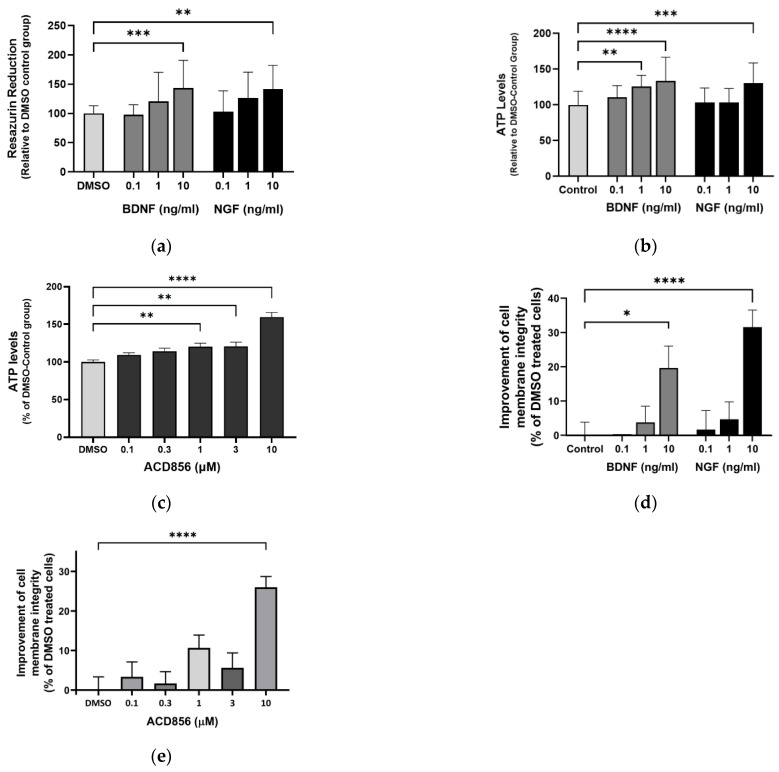
ACD856, BDNF, and NGF prevent energy-deprivation-induced stress in cortical neurons. Primary neurons were treated at DIV 4–5 with increasing doses of BDNF, NGF, and ACD856 in Neurobasal-A medium without glucose and sodium pyruvate. Metabolic activity was determined by measuring reduction in resazurin (**a**). Levels of ATP increased in the presence of BDNF and NGF (**b**) and ACD856 (**c**). Cell membrane integrity was significantly increased by BDNF and NGF (**d**) or ACD856 (**e**) in response to glucose and pyruvate withdrawal. Data shown are the means of all replicates +/− SEM from at least three different primary cultures. * *p*-value  <  0.05, ** *p*-value  <  0.01, *** *p*-value  <  0.001, **** *p*-value  <  0.0001 compared to control group at one-way ANOVA with Dunnett’s multiple comparisons test.

**Figure 9 ijms-24-11159-f009:**
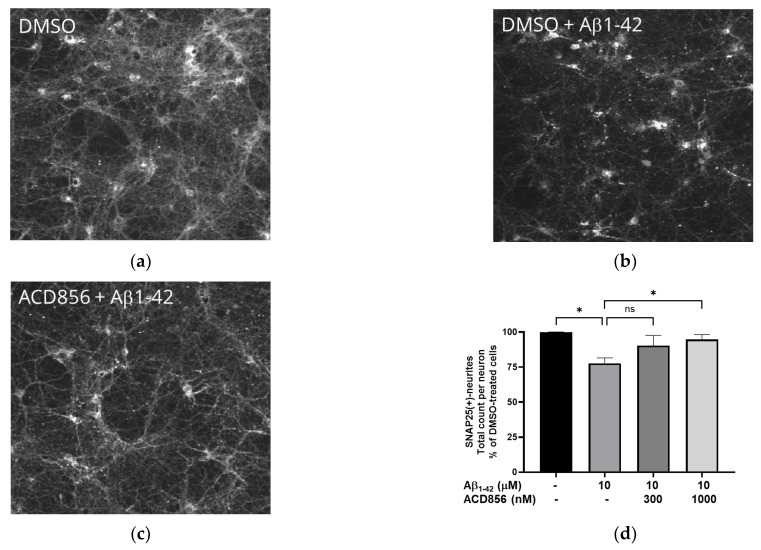
ACD856 prevented Aβ-induced decrease in SNAP25 positive neurites in cortical neurons. Primary neurons were treated at DIV 13 with vehicle (DMSO) (**a**), 10 μM Aβ_1-42_ (**b**), or 1 μM ACD856 + 10 μM Aβ_1-42_ (**c**) for 96 h in NB-medium supplemented with B27. The levels of SNAP25 were visualized and quantified by immunocytochemistry (**d**). * *p*-value  <  0.05 using one-way ANOVA with Dunnett’s multiple comparisons tests. Data shown are the means +/− SD of the average results from four independent experiments where each individual condition was performed using six different wells. The data are from four different preparations of cortical neurons. Images were acquired with X10 magnification.

**Figure 10 ijms-24-11159-f010:**
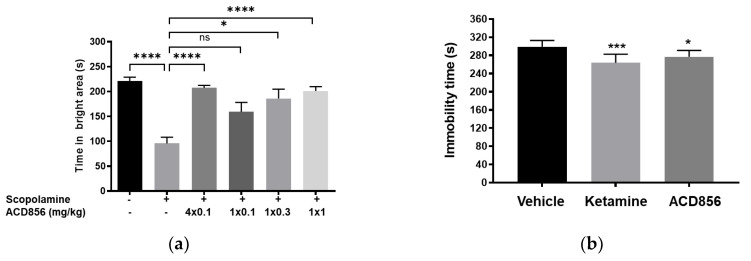
Effects of repeated administration of ACD856 on cognition and antidepressant-like behavior. ACD856 was administered s.c. for 4 consecutive days at 0.1 mg/kg or as a single s.c. dose at 0.1, 0.3, or 1 mg/kg 60 min prior to the experiment, in combination with scopolamine (0.3 mg/kg, s.c.) ACD856 reversed scopolamine-induced memory impairment at the indicated doses (**a**). * *p* < 0.05 or **** *p* < 0.0001 for scopolamine treated animals vs. the control group or ACD856-treated animals. Mice were injected for five days with a single dose of ACD856 (1 mg/kg, s.c.) per day or with a single dose of 10 mg/kg ketamine seven days prior to swimming session (**b**). Both ACD856 and Ketamine displayed prolonged antidepressive-like effects (*p* < 0.05; *p* < 0.001), significantly reducing the immobility time in the forced swim test compared with vehicle-treated mice. The bars represent the immobility time (seconds), mean ± SD (*n* = 6∓8 mice per group). The statistical analysis was performed using one-way ANOVA followed by Tukey’s test. * *p* < 0.05; *** *p* < 0.001 vs. vehicle treated group.

## Data Availability

The data presented in this study are available on request from the corresponding author. The data are not publicly available due to legal reasons.

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
