# Peer review of "Neuroprotective and Disease-Modifying Effects of the Triazinetrione ACD856, a Positive Allosteric Modulator of Trk-Receptors for the Treatment of Cognitive Dysfunction in Alzheimer’s Disease"

_ijms, 2023, doi:10.3390/ijms241311159_

Round 1
Reviewer 1 Report
The authors have conducted a preclinical study on the neurotrophic, neuroprotective and pro-cognitive function of the novel Trk positive allosteric modulator ACD856 in several models of AD in vitro and in vivo.
It is a scientifically sound, well-structured and well-written work of interest in the field of neuroscience, concerning not only AD but also, more broadly, neurodegenerative diseases, neurological disorders and aging.
However, it requires to address a few points before publication.
1) Figure 2b shows the maximum effect of ACD856 at 300 nM. This is in contrast with all the subsequent experiments in which we could see a dose-dependent effect and a higher effect even at 10uM. The authors should reason why in this particular experiment the dose-dependent response could not be observe and the lower concentration was more effective that the others.
2) Figure 2d. Why the analysis has been done by ICC and not by Western Blot? WB would have been consistent with figure 2c, as in both panels the quantity of pERK1/2 is represented. Also, the title of the y axis in fig. 2d does not look right. It mentions neurite total count which is not clearly related to pERK1/2 quantification. Additionally, the fluorescence images are required to be shown along with the graph. Authors should verify the y axis title and rephrase it to be understandable.
3) It is unclear why the first animal experiment shown in figure 6 on young mice was performed by oral administration of ACD856 daily for 5 days, while the animal experiment on old mice was performed by subcutaneous injection daily for 4 weeks? The authors should state clearly the reason of choosing the different protocols of ACD856 administration in the mice of different age. Is it possible that a more prolonged treatment of younger mice would also affect BDNF levels?
4) There is also a discrepancy in the dosage of ACD856 used in young mice (3mg/Kg) and the one used in old mice (5mg/Kg). Why is that? The two concentrations are different also compared to what reported in the materials and methods (section 4.12) which describes administrations of ACD856 at 0.1, 0.3, 1 mg/Kg in the mice for 4 days. Please check carefully the concentration and the timing of the treatment.
5) Figure 9c. Please specify in the caption the concentration of ACD856 used in the condition represented in panel c.
6) Typos: line 46. Replace BDNF and NT-4 binds with bind.
Line 131. Replace fig. 2B with fig. 2b
Figure 3a. Y axis title, replace od with of.
Line 292. Replace (fig. 9c) with (fig. 9d)
Reviewer 2 Report
The manuscript by Parrado-Fernández et al. describes the neuroprotective effect of the triazinetrione ACD856, a positive allosteric modulator of Trk-receptors, for the treatment of cognitive dysfunction in Alzheimer’s disease. Specifically, the authors provide a preclinical characterization of ACD856 demonstrating its ability to increase the levels of BDNF, to potentiate neurite outgrowth and to promote neuroplasticity adaption thus leading to improved cognitive function and long-lasting antidepressant-like effects.
The manuscript is interesting and original in its topic since it provides a new perspective in the treatment of Alzheimer disease in terms of anti-amyloid treatment. Indeed, ACD856 may fulfil many of the criteria for an adjuvant therapy or as a stand-alone treatment, if the preclinical results are reproduced in patients. However, there are some aspect of the experimental design that need to be clarified.
Major points:
The experimental design should be simplified. The authors should choose one more representative cellular model and perform the in vivo experiments in the animal model more related to the pathology. At the moment the experimental design results confusing.
1. Fig 1 and Fig 2: The authors should explain the rationale for the use of three different clonal cell line to accomplish three different information that apparently seem not tightly related to the scope of the study. Indeed, it is not clear why in figure 1 the authors measure the level of NGF in cells treated with NGF, and how they correlate this information with the effect of ACD856 on NGF production. Please clarify. The same difficult emerges in examining the results reported in figure 2 because the experimental designs described for each panel seem not related each other and even less with the experimental model of Alzheimer Disease. Please clarify.
2. In my opinion the Authors should simplify the cellular model using neurons, specifically hippocampal neurons, treated with beta amyloid, to investigate the molecular mechanisms responsible for the neuroprotective effect of ACD856.
3. To complete the analysis of the neuroprotective effect of ACD856, additional experiments should be performed evaluating mitochondrial activity in neurons treated with beta amyloid in the presence and in the absence of ACD856. These experiments would reinforce the results reported in figure 8.
4. If ACD856 is an allosteric modulator of Trk-receptros, as the authors report in the title of the manuscript, all the experiments described in the present study should be addressed to demonstrate the intracellular signalling mechanism activated downstream the receptor activation in the experimental model of interest. To this aim, primary neurons treated with ACD856 in the presence or in the absence of Beta amyloid should be assayed for BDNF and NGF production and their mediated effects, as well as for the expression of protein known to be involved in trk-recepror activation. These experiments would improve the working hypothesis and would give more relevance to the results obtained.
5. In vivo experiments in animal models showing Amyloid-beta accumulation would be useful to test the effects of ACD856 on disease progression. To this aim behavioural analysis of mice might also be useful. These experiments would be relevance to support the hypothesis that this new molecule might work as adjuvant in the therapy for AD or working as diseases modifying agent.
Minor
Please, try to be more clear in the description of the results reported in each figure.
Reviewer 3 Report
This is a very well-designed comprehensive preclinical study on the effects of the compound ACD856, a positive allosteric modulator of Trk-3 receptors. By using different in vitro models i.e recombinant cell lines U2OS-TrkA/p75-SHC1, U2OS-TrkB/p75-443 SHC1, SH-SY5Y, and PC12 cells and the primary cortical neurons as well as employing 3 weeks old C57BL/6J mice for in vivo examination, the neuroprotective and potential Alzheimer disease-modifying effects were evaluated.
The obtained results of both, in vitro and in vivo experiments, convincingly demonstrated the ACD856 positive effects on BDNF formation, neuroplasticity, and therefore on cognition, as confirmed by passive avoidance and swim-forced tests. Altogether, the results suggest ACD856 should be of benefit to dementia patients and indicate that the compound is a good candidate for further clinical studies. The discussion is multilateral, the applied methods are sound, and the relevant literature references are cited
